# Combined Spectroscopic and Computational Study of Nitrobenzene Activation on Non-Noble Metals-Based Mono- and Bimetallic Catalysts

**DOI:** 10.3390/nano11082037

**Published:** 2021-08-10

**Authors:** Reisel Millán, María Dolores Soriano, Cristina Cerdá Moreno, Mercedes Boronat, Patricia Concepción

**Affiliations:** Instituto de Tecnología Química (UPV-CSIC), Universitat Politècnica de València—Consejo Superior de Investigaciones Científicas, Avda. de los Naranjos s/n, 46022 Valencia, Spain; reimilca@itq.upv.es (R.M.); mdosorod@itq.upv.es (M.D.S.); cricermo@itq.upv.es (C.C.M.)

**Keywords:** nitrobenzene hydrogenation, IR spectroscopy, DFT, bimetallic catalysts, non-noble metals

## Abstract

In this paper, substituted anilines are industrially obtained by direct hydrogenation of nitroaromatic compounds with molecular H_2_ using metals as catalysts. Previous theoretical studies proposed that the mechanism of the reaction depends on the nature of the metal used as a catalyst, and that rationally designed bimetallic materials might show improved catalytic performance. Herein, we present IR spectroscopic studies of nitrobenzene interactions with monometallic Ni/SiO_2_, Cu/SiO_2_ and Pd/SiO_2,_ and with bimetallic CuNi/SiO_2_ and CuPd/SiO_2_ catalysts, both in the absence and presence of H_2_, combined with density functional theory (DFT) calculations on selected bimetallic NiCu(111) and PdCu(111) models. The results obtained experimentally confirm that the reaction mechanism on non-noble metals such as Ni proceeds through N-O bond dissociation, generating nitrosobenzene intermediates, while, on noble metals, such as Pd, H-attack is necessary to activate the NO bond. Moreover, a bimetallic CuPd/SiO_2_ catalyst with a Pd enriched surface is prepared that exhibits an enhanced H_2_ dissociation ability and a particular reactivity at the boundary between the two metals.

## 1. Introduction

Aniline and its derivatives are widely used in the chemical industry to produce a diversity of pharmaceuticals, agrochemicals, polymers and dyes [1,2]. Aniline is industrially obtained by direct nitrobenzene hydrogenation with H_2_ using Ni-Raney or noble metal-based heterogeneous catalysts. However, the chemoselective hydrogenation of the nitro group in substituted nitroaromatic compounds containing C=C, C=O, C≡N or halogens, which must remain unaffected during the reaction, is more challenging and requires the use of modifiers, dopants or promoters [3,4,5,6,7,8]. Alternatively, noble metal nanoparticles of Au, Ag or Pt, supported on metal oxides such as TiO_2_, Al_2_O_3_ or FeO*_x_* [9,10,11,12], Pt atomically dispersed over α-MoC [13], Pt particles or nanocrystals partly covered by TiO_2_ or Fe(OH)*_x_* [14,15,16], Pd and Pt nanoparticles dispersed on carbon or capped with organic ligands [17,18], are highly chemoselective hydrogenation catalysts. The origin of this chemoselectivity usually relies on the preferential adsorption of the nitroaromatic reactant through the nitro group, so that the other functional groups remain unreachable for hydrogenation [19,20,21].

The high cost of noble metals has led to an important research effort to develop catalysts based on cheaper and more abundant non-noble metals [8]. In recent years, promising results have been reported with Co_3_O_4_ and Fe_2_O_3_ nanoparticles stabilized on carbon and N-doped carbon supports [22,23,24], as well as with mono- [25,26,27,28,29,30] and bimetallic [31,32,33,34,35,36] catalysts containing either atomically dispersed species or nanoparticles of Ni, Co, Cu, Fe or combinations of them, supported, in most cases, on carbon-containing materials. In contrast with the large number of publications reporting catalytic data, little attention has been paid to the reaction mechanism at the molecular level, and the macroscopic scheme proposed by Haber in 1898 [37] is still used to rationalize the catalytic results. According to Haber’s mechanism (Scheme 1a), there are two main pathways to reduce the nitro group in nitrobenzene: (i) the direct route comprises three successive hydrogenation steps that convert nitrobenzene into nitrosobenzene, phenyl-hydroxylamine and, finally, aniline; (ii) in the condensation route, nitrosobenzene and phenyl-hydroxylamine intermediates react to form an azoxy compound that is transformed, by successive hydrogenation steps, into azo and hydrazo intermediates and, finally, the aniline product. Some computational studies of the reaction mechanism on noble metal [17,38,39] and non-noble metal surfaces [40,41,42] indicate that different pathways are followed on each type of catalyst (see Scheme 1b). On noble metals, such as Pt or Pd, the dissociation of the NO bonds is difficult and requires their previous transformation into weaker N–OH bonds through H attack, leading to the formation of hydroxylamine intermediates. In contrast, on non-noble metals, such as Ni or Co, the high oxophylicity of the metal surface facilitates NO bond breaking, and the reaction involves fast formation of nitrosobenzene intermediates. While this proposal is indirectly supported by analysis of the product distribution on Ni@C catalysts [41], a detailed and systematic characterization of the most relevant reaction intermediates on comparable noble and non-noble metal catalysts is still missing; this is the first motivation of the present work. In addition, a comparison of the energy profiles obtained from DFT simulations on Ni, Co, Cu and Pd surfaces showed that Cu is somewhat different from the other non-noble metals [42]. On one hand, aromatic rings do not interact strongly with the Cu(111) face; therefore, nitroaromatic compounds always adsorb in a normal orientation with respect to the metal surface, leading to a high intrinsic chemoselectivity in the case of substituted nitroaromatics. On the other hand, the activation energy barriers for the most difficult hydrogenation steps are lower on Cu(111) than in other non-noble metals, suggesting that alloying Cu with another metal able to activate H_2_ efficiently could result in a highly active and fully chemoselective catalyst [42].

In this work, monometallic Ni/SiO_2_, Cu/SiO_2_ and Pd/SiO_2_, and bimetallic CuNi/SiO_2_ and CuPd/SiO_2_ catalysts, have been prepared in the form of large metal nanoparticles on an inert support. IR studies of nitrobenzene adsorption in the absence and presence of H_2_ have been conducted to identify the key reaction intermediates, and the results have been rationalized with the assistance of DFT simulations. Our study operates from a fundamental point of view rather than catalytic one, unravelling the different reaction mechanisms on noble and non-noble metal catalysts.

## 2. Materials and Methods

### 2.1. Catalyst Synthesis and Characterization

#### 2.1.1. Catalyst Synthesis

Supported catalysts were synthesized by dissolving the corresponding amounts of nickel (II) nitrate hexahydrate, copper (II) nitrate trihydrate or palladium (II) hydrate (Sigma-Aldrich, St. Louis, MO, USA) in deionized water. Amorphous silica (200 m^2^/g) was added to this solution and was evaporated, dried overnight at 120 °C and, finally, calcined in static air at 500 °C for 2 h. Then, the solid obtained was reduced in H_2_ flow at 350 °C for 2 h. For example, in the case of monometallic samples, i.e., Ni/SiO_2_, 2.5 g of nickel (II) nitrate hexahydrate was dissolved in 40 mL of water. Then, 1.5 g of amorphous silica was added to this solution and evaporated. A green solid was obtained, which was calcined and reduced. Similar procedures have been used for the Cu/SiO_2_ and Pd/SiO_2_ samples. In the bimetallic samples, i.e., Cu_0_._5_Pd/SiO_2_, 1.9 g of copper (II) nitrate trihydrate and 0.03 g of palladium (II) hydrate was dissolved in 40 mL of water. Then, 1.5 g of amorphous silica was added to this solution and evaporated. Similar procedures have been used for the Cu_2_Pd/SiO_2_ and Cu_2_Ni/SiO_2_, but the amount of palladium (II) hydrate (0.09 g) or nickel (II) nitrate hexahydrate (0.2 g), respectively, was modified.

#### 2.1.2. Characterization

The metal content was analyzed by inductively coupled plasma optical emission spectrometry (ICP-OES) using a Varian 715-ES spectrometer.

Powder X-ray diffraction patterns (XRD) were recorded with a Philips XPert diffractometer using monochromatic Cu Kα radiation (λ = 0.15406 nm). Average particle size was calculated from the main peaks using the Scherrer equation. XRD of reduced samples was performed immediately after the ex situ reduction while expositing to air for analysis.

Temperature programmed reduction (TPR) was performed using a micromeritics Autochem 2910 apparatus with a TCD detector. The reducing gas used was 10% H_2_ in argon, with a total flow rate of 50 mL∙min^−1^ (GHSV ca. 8000 h^−1^). The temperature range explored was from room temperature to 400 °C, with a heating rate of 10 °C∙min^−1^.

X-ray photoelectron spectroscopy (XPS) measurements were performed on a SPECS spectrometer equipped with a Phoibos 150 MCD-9 detector using non-monochromatic Al Kα (1486.6 eV) X-ray radiation. The pass energy was 30 eV and the X-ray power was 100 W. Spectra treatment was performed using CASA software. Binding energies (BE) were referenced to C1s at 284.5 eV.

Hydrogen/deuterium (H_2_/D_2_) exchange experiments were carried out in a flow reactor at 25 °C. The feed gas consisted of 4 mL·min^−1^ H_2_, 4 mL·min^−1^ D_2_ and 18 mL·min^−1^ argon, and the total weight of the catalyst was 10 mg. The sample was diluted with 390 mg of SiC. Reaction products (H_2_, HD and D_2_) were analyzed with a mass spectrometer (Omnistar, Balzers). The *m*/*z* values used were 2 (H_2_), 4 (D_2_) and 3 (HD). The sample was in situ reduced at 240 °C (28 mL·min^−1^; 35.7 vol % H_2_) for 90 min with a temperature-rising rate of 10 °C·min^−1^. Then, the temperature was decreased to 25 °C and, once stabilized, the H_2_ feed was changed to the reactant gas composition. The temperature was kept at 25 °C for about 60 min.

### 2.2. IR Spectroscopy

The nature of the surface metal species and their reactivity were assessed by FTIR spectroscopy. Infrared spectra were acquired on a Bruker Vertex 70 with a DTGS detector at a resolution of 4 cm^−1^. Samples were pressed into self-supported wafers and introduced in a home-made IR cell, which was connected to both a system allowing in situ treatments and a vacuum system with a gas dosing facility.

First, samples were submitted to activation at 240 °C in H_2_ flow (10 mL·min^−1^) for 1.5 h and then evacuated under dynamic vacuum of 10^−5^ mbar at 290 °C for 1 h. Next, the temperature in the cell was decreased to room temperature prior to reactants dosing. Samples were studied in terms of nitrobenzene adsorption and its reactivity with H_2_. In the first case, nitrobenzene was dosed at room temperature until saturation (~0.3 mbar, i.e., 3.68 × 10^−6^ mol) while, in the second one, after nitrobenzene adsorption at room temperature, samples were evacuated; then, H_2_ (~12 mbar, i.e., 147 × 10^−6^ mol) was introduced to the cell. After nitrobenzene or nitrobenzene/H_2_ dosing, the temperature of the cell was increased to 60, 90, 120, 140, 160, 180 and 200 °C, and the evolution of the formed species followed at each temperature, acquiring the IR spectrum after 20 min.

For the CO titration experiments, CO was used as probe molecule to characterize the surface species of the catalyst. In this case, after sample pre-activation under similar conditions as before, the temperature of the sample was decreased to −170 °C under dynamic vacuum conditions (10^−5^ mbar). At that temperature, CO was introduced to the cell at increasing pressures from 0.1 to 9 mbar. An IR spectrum was recorded after each dose.

### 2.3. Computational Details

Periodic density functional theory (DFT) calculations were performed with the VASP code [43] using the GGA-type functional of Perdew, Burke and Ernzerhof (PBE) [44,45] and the Grimme D3 method to take into account dispersion interactions [46]. The valence density was expanded in a plane wave basis set with a kinetic energy cutoff of 450 eV, and the effect of the core electrons in the valence density was taken into account by means of a projected augmented wave (PAW) formalism [47,48]. All calculations were spin polarized. Electronic energies were converged to 10^−6^ eV and geometries were optimized by means of a conjugate gradient algorithm, until forces on atoms were smaller than 0.01 eV/Å. Transition state structures were located using the DIMER algorithm [49,50], and vibrational frequencies were calculated numerically.

The catalyst surface was simulated by means of a supercell slab model consisting of five atomic layers oriented along the (111) plane, which is the most stable and preferentially exposed in the metal catalysts considered here, and separated by a vacuum region of 10 Å to avoid interaction between periodically repeated slabs. The size of the supercell slab was also large enough to avoid interaction between the periodically repeated adsorbates.

H_2_ dissociation was investigated using a (3 × 3) supercell slab containing 45 metal atoms as catalyst model, and integration in the reciprocal space was carried out using a converged Monkhorst-Pack mesh of 6 × 6 × 1 k-points [51]. Nitrobenzene adsorption and hydrogenation were investigated using a larger 4 × 6 supercell slab containing 120 metal atoms, and integration in the reciprocal space was carried out at the Γ k-point of the Brillouin zone. During the geometry optimizations, the atomic positions of the adsorbates and the metal atoms in the two uppermost layers were allowed to fully relax, while the metal atoms in the three bottom layers were kept fixed at their bulk optimized positions. For each system, adsorption (E_ads_), activation (E_act_) and reaction (ΔE) energies were calculated as:E_ads_ = E(slab-adsorbate) − E(slab) − E(adsorbate)(1)
Eact = E(TS) − E(R)(2)
ΔE = E(P) − E(R)(3)
where E(slab-adsorbate), E(slab) and E(adsorbate) are the total energies of the optimized M(111)-adsorbate complex, clean M(111) surface model and isolated adsorbate molecule, respectively. For each elementary step considered, E(R), E(TS) and E(P) are the total energies of the corresponding reactant complex, transition state and product, respectively. Vibrational frequencies were calculated by diagonalizing the block Hessian matrix corresponding to displacements of the adsorbates. Absolute Gibbs free energies of all species were calculated as:G = E_tot_ + E_zpe_ + E_vib_ − TS_vib_(4)
where E_tot_ is the electronic energy obtained from the DFT calculation, E_zpe_ is the zero-point energy correction, E_vib_ is the vibrational thermal energy contribution and S_vib_ is the vibrational entropy. The vibrational contributions to the energy and entropy were calculated according to:(5)$$E_{\mathrm{zpe}}=\overset{3N-6}{\underset{i=1}{\sum }}\frac{1}{2}{h\upsilon }_{i}$$
(6)$$E_{\mathrm{vib}}=R\overset{3N-6}{\underset{i=1}{\sum }}\frac{{h\upsilon }_{i}}{k_{B}\left(e^{{h\upsilon }_{i}/k_{B}T}-1\right)}$$
(7)$$S_{\mathrm{vib}}=R\overset{3N-6}{\underset{i=1}{\sum }}\left[\frac{{h\upsilon }_{i}}{k_{B}T\left(e^{{h\upsilon }_{i}/k_{B}T}-1\right)}-\ln\left(1-e^{{h\upsilon }_{i}/k_{B}T}\right)\right]$$
using the vibrational frequencies ν obtained from the DFT calculations.

## 3. Results

### 3.1. Synthesis and Characterization of Monometallic Ni/SiO_2_, Cu/SiO_2_ and Pd/SiO_2_ Catalysts

In order to compare with the results from computational studies performed on extended (111) metal surface slab models, catalyst samples with high metal loadings, supported on SiO_2_ as a low dispersive support, were prepared following the procedures described in the Materials and Methods section. The metal loading determined by ICP-AES is around 23 wt% in all samples (see Table 1). The XRD patterns of the monometallic catalysts are shown in Appendix A. In the case of the calcined materials, diffraction peaks corresponding to NiO (JCPDS: 78-643), CuO (JCPDS: 48-1548) and PdO (JCPDS: 6-515) are observed in the Ni/SiO_2_, Cu/SiO_2_ and Pd/SiO_2_ samples, respectively (Appendix A). On the other hand, the XRD patterns of the monometallic catalysts after reduction (Appendix A) reveal the complete reduction of the metals showing diffraction peaks corresponding to Ni^0^ (JCPDS: 4-850), Cu^0^ (JCPDS: 4-836) and Pd^0^ (JCPDS: 87-643). The average metal particle sizes in the reduced Ni/SiO_2_, Cu/SiO_2_ and Pd/SiO_2_ samples, determined according to the Scherrer equation, are 20, 36, and 29 nm, respectively (Table 1).

### 3.2. IR Study of Nitrobenzene Hydrogenation on Monometallic Ni/SiO_2_, Cu/SiO_2_ and Pd/SiO_2_ Catalysts

With the aim of understanding the mechanism of nitrobenzene activation in the presence of non-noble metals such as Cu and Ni, compared to that of noble metals such as Pd, IR studies of nitrobenzene adsorption in the absence or presence of H_2_ were conducted on the respective monometallic supported catalysts, and the results were compared with previous DFT studies performed by our group [41,42]. The adsorption of nitrobenzene on the monometallic catalysts results in several IR bands, where the peaks at 1531 and 1350 cm^−1^ correspond to the asymmetric and symmetric stretching vibration of the nitro group, respectively, and the band at 1480 cm^−1^ corresponds to the aromatic ring [52]. The position of those peaks appears at the same frequency independently of the type of metal Ni, Cu or Pd (Appendix A), revealing a similar adsorption configuration. However, while DFT studies suggest a flat adsorption mode of the aromatic ring of nitrobenzene and the Pd surface [42], which, based on the IR selection rules, would be inactive, this may not be the case in our study, probably due to the higher coverage of adsorbed molecules favoring, in all cases, a tilted interaction between the nitro group and the metal surface. Despite this similar initial configuration, a different behavior is observed at increasing temperature depending on the type of metal. In particular, on Cu/SiO_2_ and Pd/SiO_2_ catalysts, the IR bands at 1531 and 1480 cm^−1^ decrease in intensity at increasing temperature without the formation of new IR bands, reflecting a low surface interaction in both cases (Figure 1a,b). By contrast, on the Ni/SiO_2_ catalyst, new IR bands are formed at an increasing reaction temperature concomitant with the decrease in intensity of the IR bands at 1531 and 1480 cm^−1^ (Figure 1c). Thus, at 60 and 90 °C, a shoulder at 1486 cm^−1^ is observed in the original 1480 cm^−1^ peak, which, at increasing temperature, grows in intensity, appearing as a unique peak at 180 °C. This band corresponds to nitrosobenzene [52,53], which is formed after N–O bond cleavage of the NO_2_ group on the nickel surface, and remains stable even after increasing the temperature to 200 °C. DFT studies revealed a high tendency of N–O cleavage on Ni, supporting the IR data [41,42]. Once the nitrosobenzene has been formed on the Ni/SiO_2_ catalyst, the co-addition of H_2_ at 25 °C and further increasing the temperature results in aniline formation (IR band at 1605, 1500 and 694 cm^−1^) [53] in a fast step, without the detection of intermediate species (Appendix A).

In a second part of our work, the reactivity of surface-activated nitrobenzene in the presence of H_2_ has been followed by IR studies. Different reaction trends are observed, from a very high reactivity on Pd/SiO_2_ to an intermediate one on Ni/SiO_2_, and a non-reactivity on Cu/SiO_2_ catalysts (Figure 2). In particular, in the case of the Pd/SiO_2_ catalyst (Figure 2a), aniline (IR bands at 1605 and 1500 cm^−1^) is already formed at 25 °C in the presence of both nitrobenzene and H_2_. The reactivity is almost complete, confirmed by the low intensity of the IR band of the asymmetric vibration of the nitro group at 1531 cm^−1^. In the case of the Ni/SiO_2_ catalyst (Figure 2b,c), nitrosobenzene (IR band at 1488 cm^−1^) is formed at 60 °C and, at 90 °C, aniline (IR bands at 1604, 1500 and 694 cm^−1^) starts to form. When increasing temperature from 90 to 200 °C, aniline and nitrosobenzene remain co-adsorbed, while nitrobenzene is completely transformed. Aniline is partially desorbed to the gas phase, as detected in the IR spectra after cooling down the sample (Appendix A). The conversion to the final product, i.e., aniline, is not complete, as it is in the case of the Pd/SiO_2_ catalyst. DFT calculations performed by our group [41,42] indicate a progressive blocking of active sites in Nickel-based catalysts due to O* species resulting from N–O cleavage, which are not effectively removed by H_2_. This may explain the lower formation of aniline in the Ni/SiO_2_ versus Pd/SiO_2_ catalyst. Unfortunately, the formation of Ni–O species under reaction conditions could not be detected by IR and Raman studies, being in the detection limit of both techniques. Finally, in the case of the Cu/SiO_2_ catalyst (Figure 2d), nitrobenzene hydrogenation does not take place, due to its low reactivity toward H_2_ dissociation. In fact, H_2_D_2_ isotopic experiments reveal a very low isotopic HD exchange in the case of the Cu/SiO_2_ catalyst (Table 1).

In the second part of our study, bimetallic catalysts are considered in an attempt to promote the reactivity of the copper-based catalyst. As such, bimetallic CuNi/SiO_2_ and CuPd/SiO_2_ systems are studied.

### 3.3. Synthesis and Characterization of Bimetallic CuNi and CuPd Catalysts

The bimetallic samples have been labelled as CuXMe (Me = Pd or Ni), where X represents the weight percent of the second element (see Table 2). Based on the low metal loading of the second element, only diffraction peaks corresponding to CuO (JCPDS: 48-1548) are observed in the XRD pattern of the calcined Cu_0_._5_Pd/SiO_2_ and Cu_2_Pd/SiO_2_ (Appendix A). However, in the Cu_2_Ni/SiO_2_ sample, in addition to the diffraction peaks corresponding to CuO, diffraction peaks corresponding to NiO (JCPDS: 78-643) are also observed as a minoritary phase. In the reduced samples, diffraction peaks corresponding to Cu^0^ (JCPDS: 4-836) are observed in all samples (Appendix A). The average copper particle sizes in the reduced Cu_0_._5_Pd/SiO_2_, Cu_2_Pd/SiO_2_ and Cu_2_Ni/SiO_2_ samples, determined according to the Scherrer equation, are ~15 nm in the samples with Pd, and ~30 nm in the Ni sample, similar to that of the monometallic Cu/SiO_2_ catalyst (Table 2).

The interaction between both metals can be envisaged by a higher reducibility of the bimetallic catalysts compared to the reference monometallic Cu/SiO_2_ catalysts, as visualized from the TPR-H_2_ pattern (Appendix A). The surface distribution of both metals in the bimetallic samples has been studied by X-ray photoelectron spectroscopy (XPS) and Infrared spectroscopy (IR) using CO as a probe molecule. The XPS data of the Cu_0_._5_Pd/SiO_2_, Cu_2_Pd/SiO_2_ samples (Appendix A) show, in both cases, the same Pd/Cu atomic ratio (0.2), whereas, IR of CO, which is more surface sensitive, shows a Pd rich surface in the Cu_2_Pd/SiO_2_ sample; a Cu rich surface is observed in the Cu_0_._5_Pd/SiO_2_ sample (Appendix A). In the Cu_2_Ni/SiO_2_, while the surface composition determined from XPS analysis is higher than in the previous samples (Ni/Cu atomic ratio = 0.9), the IR-CO shows IR peaks predominantly associated with Cu. Here, XPS has a penetration depth of around 6 nm, while IR-CO is sensitive to the upper layer exclusively. On the other hand, while H_2_ activation is important for hydrogenation activity, H_2_D_2_ isotopic studies have been conducted, revealing, in all bimetallic samples, a markedly higher H_2_ activation than in the monometallic Cu/SiO_2_ one. In fact, the HD formation increases in the order Cu/SiO_2_ << Cu_0_._5_Pd/SiO_2_ < Cu_2_Ni/SiO_2_ < Cu_2_Pd/SiO_2_ (Table 1 and Table 2).

### 3.4. IR Study of Nitrobenzene Hydrogenation on Bimetallic CuNi and CuPd Catalysts

The adsorption of nitrobenzene on the Cu_0_._5_Pd/SiO_2_, Cu_2_Pd/SiO_2_ and Cu_2_Ni/SiO_2_ catalysts, and their evolution at increasing temperatures, is shown in Figure 3.

In particular, in the Cu_0_._5_Pd/SiO_2_ catalyst, the interaction of nitrobenzene with the catalyst surface is low, analogous to that of the monometallic Cu/SiO_2_ catalyst, desorbing when increasing the reaction temperature from 60 to 180 °C (Figure 3a). In fact, IR-CO reveal a small amount of Pd on the catalyst’s upper surface, as the surface is enriched with Cu. When increasing the Pd loading in the catalyst (Cu_2_Pd/SiO_2_), a higher reactivity of nitrobenzene on the catalyst surface is observed, consistent with the Pd enrichment on the upper surface detected by IR-CO. As displayed in Figure 3b, nitrobenzene adsorption leads to the formation of nitrosobenzene (IR band at 1485 cm^−1^) at 60 °C, achieving a maxima peak intensity at 120 °C, in parallel with the consumption of nitrobenzene. A further increase in the temperature up to 140 and 180 °C results in the disappearance of the band associated with nitrosobenzene and the appearance of new IR bands at 642 and 2199 cm^−1^ (Figure 3b,c), which may be ascribed to Pd–O and/or Pd–N interaction [54]. These bands may arise from a second N–O cleavage of nitrosobenzene, in agreement with the mechanism reported in the DFT studies (see next section). Notoriously, despite the fact that the surface is enriched in Pd, which is a noble metal, the reaction mechanism follows a different path. Thus, a N–O dissociative reaction path, as in the case of non-noble metal catalysts, is observed, in contrast to the NO_2_-hydrogenation path observed in noble metals. Next, after co-addition of H_2_ at 25 °C and an increase in temperature to 200 °C, the previously reported IR bands at 642 and 2199 cm^−1^ are hydrogenated (not shown).

In the presence of Ni, the behavior of the Cu_2_Ni/SiO_2_ catalyst is different from the Cu_2_Pd/SiO_2_, but similar to that of Cu/SiO_2_ (Figure 3d), on which nitrobenzene desorption is favored. This is not surprising, based on the low surface distribution of Ni species as detected by IR-CO.

Finally, the hydrogenation of nitrobenzene in the presence of nitrobenzene and H_2_ has been studied by IR on the most active Cu_2_Pd/SiO_2_ catalyst (Figure 4), displaying a high reactivity toward aniline formation. Thus at 60 °C, aniline and nitrosobenzene are observed, with nitrosobenzene being completely hydrogenated to aniline at 140 °C, with practically 100% selectivity to aniline at this temperature. The reactivity of this sample is lower than that of the pure Pd/SiO_2_ catalyst and, as in that case, no intermediate species, such as hydroxylamine, are observed.

### 3.5. DFT Study of Nitrobenzene Adsorption and Reaction on Mono- and Bimetallic Catalysts

As described in previous studies [41,42], the main elementary steps in the mechanism of nitrobenzene hydrogenation to aniline catalyzed by non-noble metals are those summarized in Equations (8)–(13):H_2_ → 2H*(8)
Ph–NO_2_ → Ph–NO_2_*(9)
Ph–NO_2_* → Ph–NO* + O*(10)
Ph–NO* → Ph–N* + O*(11)
Ph–N* + H* → Ph–NH*(12)
Ph–NH* + H* → Ph–NH_2_*(13)

H_2_ dissociation into two adsorbed H atoms, according to (8), involves low or null activation barriers on Ni(111) and Pd(111), but it is more difficult on a Cu(111) surface because H_2_ does not adsorb on the metal but reacts from the gas phase with a calculated activation energy of 8.5 kcal/mol. [42] However, the addition of just a single Ni atom on a perfect Cu(111) surface, as in the Ni_1_Cu(111) slab model depicted in Figure 5, is enough to bind H_2_ with a calculated adsorption energy of −9.3 kcal/mol and dissociate it with a slightly lower barrier of 7.8 kcal/mol. Then, after nitrobenzene adsorption on the metal surface according to Equation (9), one of the two NO bonds of the nitro group is easily broken, generating nitrosobenzene and one surface oxygen atom, (see Equation (10)). The subsequent dissociation of the NO bond in nitrosobenzene, Equation (11), yields a highly stable Ph–N* intermediate with one N atom tri-coordinated to the surface metal atoms. The activation energies for these two NO bond breaking steps, summarized in Table 3, are similar in the three models considered, and always equal or lower than 10 kcal/mol. However, the interaction between the adsorbed species and the metal surface is clearly stronger in the case of Ni(111), as indicated by the more exothermic reaction energy values listed in Table 3 and the energy profiles compared in Figure 5. Interestingly, the stability of the Ph–N* intermediate is modified by the presence of just one Ni atom on the Cu(111) surface, and the calculated reaction energy for step (11) decreases by 8 kcal/mol in Ni_1_Cu/(111) compared with Cu(111). The lower stability of Ph–N* intermediate facilitates its subsequent hydrogenation according to (12) and (13) to finally yield adsorbed aniline. While these results would suggest a better catalytic performance of the bimetallic Cu_2_Ni/SiO_2_ catalyst when compared to Cu/SiO_2_, the IR study described above shows that neither Cu/SiO_2_ nor Cu_2_Ni/SiO_2_ are able to activate nitrobenzene, which desorbs when temperature is raised to 200 °C. The same behavior is observed with the Cu_0_._5_Pd/SiO_2_ sample containing a low amount of Pd on the catalyst surface, but not with the Cu_2_Pd/SiO_2_ sample, in which small surface Pd regions are detected by IR-CO.

At this point, it becomes necessary to study the adsorption of nitrobenzene and its evolution with temperature on catalyst models able to accurately represent the surface composition of the mono- and bimetallic samples experimentally tested. To this end, the Gibbs free energies of nitrobenzene adsorption were calculated at increasing temperatures for Cu(111), Ni_1_Cu/(111), Pd(111), and a series of Pd_n_Cu(111) slab models were generated by replacing an increasing amount of Cu atoms with Pd_n_, n = 1 to n = 10 (see Figure 6 and Figure 7). As previously reported [42], nitrobenzene adsorbs normally to the Cu(111) surface with each O atom of the nitro group directly bonded to one surface Cu atom. In contrast, nitrobenzene adsorbs parallel to the Pd(111) surface, with the C atoms of the aromatic ring strongly bonded to the surface Pd atoms, and with the nitro group tilted with only one O atom interacting with the metal. The most stable structure for nitrobenzene adsorption on Pd_1_Cu(111) is similar to that found for Ni_1_Cu/(111), i.e., normal to the surface and with one of the O atoms of the nitro group directly bonded to the isolated Ni or Pd atom. Introduction of Pd atoms in the top layer of the catalyst, as in Pd_5_Cu(111) model (see Figure 6), displaces adsorbed nitrobenzene in such a way that the two O atoms of the nitro group are directly bonded to two Cu atoms, but the molecule is partly bent due to some interaction between the aromatic ring and the surface Pd atoms. The arrangement of the Pd atoms in Pd_6_Cu(111), Pd_8_Cu(111) and Pd_10_Cu(111) models, occupying positions in the two uppermost layers of the model, enhances the interaction through the aromatic ring and results in de-coordination of the nitro group from the catalyst surface, with adsorption geometries quite similar to that found on pure Pd(111) surface. These geometrical changes are clearly reflected in the calculated Gibbs free adsorption energies plotted in Figure 7. Nitrobenzene interaction with Cu(111) is weak, and desorption could be spontaneous when raising the temperature. The addition of isolated Ni or Pd atoms slightly modifies this interaction, enhancing it in the first case (dotted orange line in Figure 7) and weakening it in the second one (yellow dotted line in Figure 7). In agreement with this trend, the normal adsorption of nitrobenzene on Ni(111) is more stable than on Cu(111) and on Ni_1_Cu/(111), despite the equivalent geometry. The destabilizing effect associated with Pd is reversed as the number of Pd atoms increases and the nature of the interaction changes. Thus, the Gibbs free energy of adsorption on Pd_5_Cu(111) model (red dotted line) is equivalent to that on Cu(111) but, when the interaction through the aromatic ring becomes predominant, the system is significantly stabilized, and no spontaneous desorption is expected in the range of considered temperatures (purple and blue dotted lines in Figure 7).

These data are fully consistent with the experimental results found for mono-metallic Cu/SiO_2_, Ni/SiO_2_ and Pd/SiO_2_ catalysts. The two non-noble metal based catalysts should be able to activate the nitro group in the absence of H_2_, but the weak adsorption of nitrobenzene on Cu(111) leads to its spontaneous desorption before bond-breaking occurs. The higher stability of nitrobenzene adsorbed on Ni(111) makes possible the subsequent reaction following the proposed pathway via NO bond dissociation, and nitrosobenzene is observed by IR spectroscopy at 60 °C. After the addition of H_2_, it is necessary to raise the temperature to 90 °C to detect aniline, in agreement with the higher activation barriers calculated for the hydrogen transfer steps (see Table 3). In contrast, Pd behaves as a noble metal and is not able to dissociate the NO bond in the absence of H_2_. However, aniline is already detected at 25 °C when the experiments are carried out in the presence of both nitrobenzene and H_2_, confirming the different mechanism operating on Pd/SiO_2_ catalysts.

The experimental results obtained with the bimetallic catalysts are also explained by the theoretical data. Too dispersed Ni or Pd species on the Cu(111) surface do not sufficiently stabilize adsorbed nitrobenzene, and therefore the behavior of Cu_2_Ni/SiO_2_ and Cu_0_._5_Pd/SiO_2_ samples is similar to that of Cu/SiO_2_. However, the small agglomerates of Pd atoms formed on the surface of the Cu_2_Pd/SiO_2_ sample are able to bind nitrobenzene and make it react, both in absence and presence of H_2_. Indeed, the two pathways described in Scheme 1b for noble and non-noble metals are accessible in this system. Starting from nitrobenzene adsorbed on the Pd_10_Cu(111) model, we calculated the transition states for both NO dissociation to give nitrosobenzene (TS-PhNO in Figure 6) and for H transfer to form adsorbed Ph–NOOH (TS-PhNOOH in Figure 6). The calculated activation energies listed in Table 4 indicate that H transfer is preferred in this bimetallic system with a barrier of 9 kcal/mol, but formation of nitrosobenzene with a calculated barrier of 19.3 kcal/mol is also energetically accessible. Notice that the calculated E_act_ values for the hydrogen transfer steps to Ph–N* and Ph–NH* intermediates on Ni(111) are larger (26.1 and 19.6 kcal/mol, see Table 4), and the E_act_ value for the first hydrogenation step to form Ph-NOOH* on Pd(111) is quite close, 15.9 kcal/mol (see Table 4). These values are in excellent agreement with the observation of nitrosobenzene on the Cu_2_Pd/SiO_2_ sample, both in the absence and presence of H_2_ and with the fast formation of aniline in the presence of H_2_.

## 4. Conclusions

Based on previous theoretical studies proposing that nitrobenzene hydrogenation on noble and non-noble metals follows different pathways, in this work, mono- and bimetallic catalysts based on Ni, Cu and Pd have been synthesized, characterized and their interaction with nitrobenzene and H_2_ has been investigated by IR spectroscopy and DFT calculations. First, it has been confirmed by IR spectroscopy that nitrobenzene is directly converted into nitrosobenzene on large Ni particles supported on SiO_2_, and that the addition or co-adsorption of H_2_ results in the formation of aniline, following the mechanism proposed theoretically in ref. [41]. In contrast, neither Cu nor Pd are able to activate nitrobenzene in the absence of H_2_. Co-adsorption of nitrobenzene and H_2_ on Pd/SiO_2_ leads to aniline without detection of nitosobenzene, in agreement with the pathway proposed for noble metals, while Cu/SiO_2_ is not active due to its inability to dissociate H_2_ [42], as confirmed by H_2_D_2_ isotopic studies. Second, with the aim of improving the H_2_ dissociation ability of Cu/SiO_2_, bimetallic catalysts based on Cu doped with different amounts of either Ni or Pd were prepared, characterized by XPS and IR of CO adsorption and their interaction with nitrobenzene was followed by IR spectroscopy. When the amount of doping metal on the surface of the Cu nanoparticles is low, the bimetallic catalysts behave as Cu/SiO_2_, irrespective of the nature of the doping metal, either Ni or Pd. However, the bimetallic Cu_2_Pd/SiO_2_ catalyst containing a Pd-enriched surface exhibits not only enhanced H_2_ activation but also different reactivity. Thus, interaction of Cu_2_Pd/SiO_2_ with nitrobenzene alone results in the formation of nitrosobenzene according to the N–O dissociation mechanism proposed for non-noble metals, while co-adsorption of H_2_ and nitrobenzene leads to fast hydrogenation following the H-transfer pathway preferred on noble metals.

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
