# Peer review of "Combined Spectroscopic and Computational Study of Nitrobenzene Activation on Non-Noble Metals-Based Mono- and Bimetallic Catalysts"

_nanomaterials, 2021, doi:10.3390/nano11082037_

Round 1

Reviewer 1 Report

Nanomaterials 1330933

Combined Spectroscopic and Computational Study of Nitro- benzene Activation on Non-Noble Metals based Mono- and Bi- metallic Catalysts

Authors: Reisel Millán, María Dolores Soriano , Cristina Cerdá Moreno , Mercedes Boronat , and Patricia Concepción

The authors use Ir spectroscopy to study mono and bimetallic catalysts for the hydrogenation of nitrobenzene combined with DFT calculations. They have found that the reaction mechanism on non-noble metals like Ni proceeds through N-O bond dissociation generating nitrosobenzene intermediates, while on noble  metals like Pd, H-attack is necessary to activate the NO bond. Finally, Moreover, CuPd/SiO2 catalyst with a Pd enriched surface exhibits an enhanced H2 dissociation ability and a particular reactivity at the boundary between the two metals.

The work is interesting and easy to follow. It can be accepted for publication in Nanomaterials. My only comment is why the authors used bimetallic catalysts with Cu as base monometallic catalyst. From their results I believe that they should also check the NiPd bimetallic catalyst since the two monometallic catalysts exhibit significant differences in the mechanism and the possible synergy may be interesting. Also, why they did not use a higher amount of Pd to determine the best surface concentration?

Some other minor points are:

The authors should discuss better the TPR results

Why the surface ratio is the same for the two Cu/Pd catalysts? If the ratio is the same why the two catalysts have different activity?

In scheme 1 the authors should include the noble or non-noble metals to help the readers with the mechanism. Strictly speaking it is nickel(II) not nickel (II) (without space) etc . please  check and correct in several places in manuscript. Also it is mL not ml

Please give a table with the characteristics of the catalysts. (for example content, SSA etc) in text and not in SI

The authors should determine the activity of each catalyst if they want to discuss what catalyst is more active. Otherwise they should rephrases the text in manuscript. With their interesting work they cannot provide data about activity, (conversion or rate)

Reviewer 2 Report

In this manuscript, the authors performed IR spectroscopy experiments and density functional theory calculations to understand the bonding structure of nitro-benzene on Pd, CuNi, and CuPd surfaces. The experiments are performed carefully, and the computation gives support to understand the results. However, I have a few questions before accepting for publication.

First, it is mentioned that the metal bound oxygen such as Ni-O was not detected in their IR experiments. Why is this? Is this due to the weak adsorption intensity, or just a concentration issue?

Next, the authors performed the calculation of different concentrations of Pd on the surface in Figure 6, but how were these structures generated. Also, if one fixes the Pd number and varies the geometry, how much more stable is the geometry that they used, which was clustering the Pd atoms together compared to geometries with two islands of Pd clusters or randomly dispersed single atom Pd islands?

Lastly, the authors relate the reactivity to the adsorption of the reactant nitrobenzene, but it is also important that the product is desorbed from the catalyst's surface. Is there any experimental proof of the desorbed aniline in the gas phase? Also, how much is the calculated desorption energy of aniline? I think the authors can estimate the desorption rate from this energy.

Reviewer 3 Report

The manuscript by Millán et al describes the combined experimental and theoretical study of the non-noble (Ni, Cu) metals, as well as Pd, adsorbed on SiO2 support for catalytic hydrogenation of nitrobenzene. The motivation for this work is clear. The introduction sufficiently describes the background of the research field. The catalytic performance was monitored in situ by IR spectroscopy. The theoretical DFT calculations constitute significant part of the work. The manuscript is, in general, accurate, well written and can be recommended for publication after addressing the following minor problems:

1) Please clarify the “PhN” and all other species in the Scheme 1b by adding more data regarding the electronic configuration of the intermediates (e.g. charge, radicals where applicable, lone pairs etc). Further, I found only  limited data on the respective mechanism (highlighted by red rectangle) in the refs. [40-42], which concern the non-noble metal catalysis, while these references are cited in the line 55 as those showing the Scheme 1b mechanism. Please check and correct.

2) Please add more data on the real conditions of the catalytic studies, i.e. loadings (g) of the M/SiO2 samples as well as absolute amounts of nitrobenzene, at least the ones expected from the pressure and reactor volume. This would help the readers to compare the presented results with their own.  

3) The Authors wrote in the line 236: “Different reaction rates”. Although one could suppose that the respective rates really differ, since the Authors did not perform the kinetic studies (IR data were taken at 20 min) the use of the word “rate” should be avoided, replacing it in with a more appropriate term which reflect the experimental data (e.g. “yield” or “amount”).

Author Response

"please see the attachment"

Round 2

Reviewer 1 Report

The authors have revised their manuscript according to my suggestions.

Their work can be accepted for publication

Please correct * with × in several points in text